Different gut microbial types were found in captive striped hamsters

Fan Chao fanchao@qfnu.edu.cn
Zheng Yunjiao
Xue Huiliang
Xu Jinhui
Wu Ming
Chen Lei
Xu Laixiang xulx@qfnu.edu.cn
School of Life Sciences, Qufu Normal University , Qufu , Shandong , China
Syed Mudasir Ahmad
Electronic publication date: 2023 Nov 6
Publication date: 2023
Volume: 11
Electronic Location ID: e16365
Received 2023 Apr 6; Accepted 2023 Oct 6
Copyright: ©2023 Fan et al.
Copyright year: 2023
Copyright holder: Fan et al.
License: This is an open access article distributed under the terms of the Creative Commons Attribution License, which permits unrestricted use, distribution, reproduction and adaptation in any medium and for any purpose provided that it is properly attributed. For attribution, the original author(s), title, publication source (PeerJ) and either DOI or URL of the article must be cited.
License URL: https://creativecommons.org/licenses/by/4.0/

Keywords: Gut microbiota, Rodent, 16S rRNA, Typing analysis, Sex

Funding: National Natural Science Foundation of China 32072436 31972283 Initial Scientific Research Fund for Young Teachers at Qufu Normal University 612501 This work was supported by the National Natural Science Foundation of China (No. 32072436 and No. 31972283) and the Initial Scientific Research Fund for Young Teachers at Qufu Normal University (612501). The funders had no role in study design, data collection and analysis, decision to publish, or preparation of the manuscript.

==============================
Background

Typing analysis has become a popular approach to categorize individual differences in studies of animal gut microbial communities. However, previous definitions of gut microbial types were more understood as a passive reaction process to different external interferences, as most studies involve diverse environmental variables. We wondered whether distinct gut microbial types can also occur in animals under the same external environment. Moreover, the role of host sex in shaping gut microbiota has been widely reported; thus, the current study preliminarily explores the effects of sex on potential different microbial types.

Methods

Here, adult striped hamsters Cricetulus barabensis of different sexes were housed under the same controlled laboratory conditions, and their fecal samples were collected after two months to assess the gut microbiota by 16S rRNA sequencing.

Results

The gut microbiota of captive striped hamsters naturally separated into two types at the amplicon sequence variant (ASV) level. There was a significant difference in the Shannon index among these two types. A receiver operating characteristic (ROC) curve showed that the top 30 ASVs could effectively distinguish each type. Linear discriminant analysis of effect size (LEfSe) showed enrichment of the genera Lactobacillus, Treponema and Pygmaiobacter in one gut microbial type and enrichment of the genera Turicibacter and Ruminiclostridium in the other. The former type had higher carbohydrate metabolism ability, while the latter harbored a more complex co-occurrence network and higher amino acid metabolism ability. The gut microbial types were not associated with sex; however, we did find sex differences in the relative abundances of certain bacterial taxa, including some type-specific sex variations.

Conclusions

Although captive animals live in a unified environment, their gut bacteria can still differentiate into distinct types, but the sex of the hosts may not play an important role in the typing process of small-scale captive animal communities. The relevant driving factors as well as other potential types need to be further investigated to better understand host-microbe interactions.

Introduction

Trillions of microbes inhabit the guts of animals and perform vital functions for their hosts (Voigt et al., 2015; Bo et al., 2019). Studies on population stratification have indicated that considerable variation in gut microbial composition is very common among individuals of the same population (David et al., 2014; Maurice et al., 2015). To better generalize the individual differences in symbiotic microbiota, typing analysis has been widely applied in relevant studies (Willis et al., 2018; Cheng & Ning, 2019; Liu et al., 2022; Hu et al., 2023). Researchers first clustered the human gut microbiome into different types, which have been described as “densely populated areas in a multidimensional space of microbial community composition” (Arumugam et al., 2011; Ding & Schloss, 2014). Subsequently, gut microbial types have been found in several species, such as mice, gorillas, rats, pikas and bees (Hildebrand et al., 2013; Moeller et al., 2015; Zhang et al., 2019; Yu, Li & Li, 2022; Hu et al., 2023). These typing analyses divide the gut microbial community into substructures at the genus or operational taxonomic unit (OTU) level and label them enterotypes, community types or enterotype-like clusters to represent the states of symbiotic bacterial composition and different local optima in gut community effectiveness.

However, differences in gut microbiota can be largely determined by the living conditions of the hosts. Therefore, the gut microbial types occurring in humans and wild animals are more likely driven by external factors and can be regarded as the adaptive characteristics of gut microbiota to different environments. For example, the Bacteroides-dominant type in humans is strongly associated with a diet rich in animal proteins and saturated fats, whereas the Prevotella-dominant type is typical of a fiber-rich diet (Arumugam et al., 2011; David et al., 2014). Studies on wild mice and Tibetan wild asses (Equus kiang) also provided strong evidence that food resources highly contribute to the presence of gut microbial types (Wang et al., 2014; Liu et al., 2022), while the two gut microbial types of wild plateau pikas (Ochotona curzoniae) had visibly altitude-associated distributions (Yu, Li & Li, 2022). Gut microbial types presented under the experimental environments were also likely to be adaptive variations to stresses, as they were grouping-specific and shaped by disease model, medication or other special treatments (Hildebrand et al., 2013; Zhang et al., 2019). Therefore, can different gut microbial types also occur in animal hosts under the same living conditions?

The effect of host sex on shaping gut bacteria has become the focus of microbial studies (Sisk-Hackworth, Kelley & Thackray, 2023). Sex bias of the composition of the gut microbiome in mice has been observed (Org et al., 2016), and similar conclusions have also been presented in studies of humans and rats (Jaggar et al., 2020; Gu et al., 2021). The differences in enrichment of bacterial taxa and the separation of male and female samples are usually related to sexual dimorphism in host physiological traits and are of great importance to the “gut-brain” axis (Takagi et al., 2019; He et al., 2021; Snigdha et al., 2022). In addition, sex-specific differences are affected by the host’s living environment; for example, the compositional and functional variations in the gut microbiota of male and female red deer (Cervus elaphus) presented inconsistencies among wild and captive conditions (Sun et al., 2023). It is also worth noting that the gut microbiota of male and female mammals will have different responses to the same external factors, especially diet (Bolnick et al., 2014). On this basis, we further wondered whether sex plays a role in the formation of gut microbial types in captive animals and compared the sex-specific differences in gut microbiota under the same controlled environment.

The striped hamster (Cricetulus barabensis) is a small mammal widely distributed in the temperate zone of East Asia, especially in northern China, and is one of the main rodent pests in farmland that has high reproductive capacity; thus, it has received much attention from ecology researchers and rodent control personnel (Xue et al., 2021). Recent studies on this rodent are still mainly focused on individual physiology (Xue et al., 2021; Xue et al., 2022), and knowledge regarding its gut microbiome is limited. In this study, striped hamsters were used as the experimental animals and housed under the same laboratory conditions. Their feces were collected for high-throughput 16S rRNA gene sequencing to conduct gut microbial typing analysis.

Materials & Methods

Sample collection

Twenty-four striped hamsters (male: eight, female: 16) were captured by the live-trap method from the Qufu region of Shandong Province, China. Then, the experimental animals were taken to the animal feeding room of Qufu Normal University and maintained for two months before the sampling date. Each hamster was individually reared in an opaque plastic box to prevent microbial transfer between the animals. A controlled laboratory environment under natural light with an ambient temperature of 22 ± 2 °C and a relative humidity of 55% ± 5% was provided to hamsters. Water and the same artificial rodent feed (Qianmin Feed, Shenyang, Liaoning, China) were provided ad libitum. All sampled individuals were adults with a body weight greater than 20 g. The collection of fecal samples took place within one day during daylight hours to minimize any potential influence from the circadian rhythm. Each individual was kept in a cage sterilized beforehand using 75% alcohol, and fecal samples were collected in 2-mL cryogenic vials (Corning, Reynosa, TAMP, Mexico) within 1 min of defecation. The collected feces were placed in liquid nitrogen and then stored in a −80 °C ultralow temperature freezer (Thermo Fisher Scientific, Waltham, MA, USA).

All procedures have followed the Laboratory Animal Guidelines for the Ethical Review of Animal Welfare (GB/T 35892–2018) and have been approved by the Biomedical Ethics Committee of Qufu Normal University (Permit Number: dwsc2023005).

DNA extraction and sequencing

Microbial genomic DNA was extracted from fecal samples by using the MJ-Feces DNA Kit (Majorbio, Shanghai, China) according to its manufacturer’s protocol. The quality of all DNA samples was verified, and their concentration was measured using NanoDrop 2000 spectrophotometers (Thermo Fisher, Wilmington, DE, USA). The hypervariable region V3-V4 of the bacterial 16S rRNA gene fragments were amplified from the extracted DNA using common primers 338F (5′-ACTCCTACGGGAGGCAGCAG-3′) and 806R (5′-GGACTACHVGGGTWTCTAAT-3′) (Caporaso et al., 2011). The polymerase chain reaction (PCR) amplification was performed as follows for 27 cycles: denaturing at 95 °C for 30 s, annealing at 55 °C for 30 s, and extension at 72 °C for 45 s. The PCR mixtures contained 5 × TransStart FastPfu buffer (4 µL), 5 µM each primer (0.8 µL), 2.5 mM deoxynucleoside triphosphates (dNTPs) (2 µL), extracted DNA (10 ng), TransStart FastPfu DNA Polymerase (0.4 µL), and ddH2O to make the total volume 20 µL. The size of amplicons was confirmed through the utilization of agarose gel electrophoresis. Amplicons were subjected to paired-end sequencing on the Illumina MiSeq sequencing platform using PE300 chemical at Majorbio Bio-Pharm Technology Co. Ltd. (Shanghai, China). The raw reads have been uploaded to the National Center for Biotechnology Information database under the accession number PRJNA937404, and sex information can be found in the Mapping File S1.

Bioinformatics and statistical analysis

The resulting sequences obtained after demultiplexing were merged with FLASH (v1.2.11) and quality filtered with fastp (v0.19.6) based on these parameters (Magoč & Salzberg, 2011; Chen et al., 2018): (1) the reads were truncated receiving an average quality score <20 over a 50 bp sliding window, and the truncated reads shorter than 50 bp, as well as reads containing ambiguous characters, were all filtered; (2) paired reads were merged when overlapping sequences longer than 10 bp and the maximum mismatch ratio of the overlap region was 0.2, while reads that could not be assembled were discarded; (3) samples were distinguished according to the barcode and primers, and the sequence direction was adjusted, with exact barcode matching and two nucleotide mismatches in primer matching. The remaining sequences were denoised using the Divisive Amplicon Denoising Algorithm 2 plugin implemented in QIIME2 software (v2020.2) by filtering out noise (Callahan et al., 2016), and then the denoised sequences called amplicon sequence variants (ASVs) were obtained. ASV taxonomic assignment was conducted by the naive Bayes consensus taxonomy classifier integrated in the QIIME2 pipeline based on the SILVA bacterial 16S rRNA database (v138). The ASVs belonging to chloroplasts, mitochondria, and archaea were filtered before performing downstream analysis. To minimize the influence of different sequencing depths, normalization was performed according to the minimum value of sequence counts among all fecal samples, and the number of sequences in every single sample has been rarefied to 15,594, which still had a Good’s coverage of more than 99.9%. The ASV table has been uploaded as a supplemental material. Alpha diversity indices calculated by Mothur (v1.30) were used to evaluate gut microbial richness and diversity. Bacterial function was calculated by PICRUSt2 software against the Kyoto Encyclopedia of Genes and Genomes (KEGG) database. After obtaining the KO (KEGG Orthology) information, the abundance tables of functional pathways at three different levels were obtained by comparing KOs with the KEGG pathway database (Douglas et al., 2020).

Statistical analysis was mainly performed using integrated R software in the majorbio cloud platform (https://cloud.majorbio.com/). The optimal number of types was chosen based on Calinski–Harabasz (CH) values calculated by the R package “clusterSim”, and clustering of the gut microbial types was performed by the partitioning around medoid (PAM) method based on the Jensen–Shannon dissimilarity of ASVs calculated by the R package “cluster” (https://enterotype.embl.de/enterotypes.html). The Mann–Whitney U test was applied to detect differences in the relative abundances of ASVs and alpha diversity indices. Principal coordinate analysis (PCoA) based on beta diversity and Adonis analysis (PERMANOVA, permutational multivariate analysis of variance) were performed using the “vegan”, “ade4” and “ggplot2” packages. Linear discriminant analysis effect size (LEfSe) was used to identify the microbial taxa that most contributed to the differences between groups. The pairwise correlation test among ASVs was determined by the R package “stat”, and it was performed by screening out significant Spearman’s correlations with absolute coefficient values greater than 0.6, followed by using Gephi software (v0.9.2) to draw the co-occurrence networks. Fisher’s exact test was used to analyze the influence of sex on the classification of gut microbial types. A Venn diagram was used to display the distribution of ASVs among different types. The receiver operating characteristic (ROC) curve based on the relative abundances of the top 30 ASVs was calculated by the R package “pROC”. Functional differences in level-2 KEGG pathways were identified and visualized using STAMP v2.1.3 software.

Results

Typing analysis of the gut microbiota in captive striped hamsters

High-quality sequences of the 16S rRNA gene from all the fecal samples were assigned into 1946 ASVs after data processing. The rarefaction curves showed that as the number of sampled reads increased, the number of observed ASVs gradually stabilized, and there was no further growth or fluctuation (Fig. S1). The highest CH index value was determined to be two (Fig. S2), and Fig. 1A displays the two gut microbial types. Among the 24 samples, 14 (58.3%) were assigned to Type 1, and 10 (41.7%) were assigned to Type 2. A Venn diagram showed that there were 523 ASVs unique to Type 1, while 519 ASVs were unique to Type 2, and 904 ASVs were shared between the two types (Fig. 1B). Although there was no significant difference in the Sobs index (p = 0.107) and PD index (p = 0.135) between them (Figs. 1C, 1D), Type 1 had a higher Shannon index (p = 0.009) than Type 2 (Fig. 1E).

Figure 1 Identification of gut microbial types in captive striped hamsters.

(A) The two gut microbial types clustered at the ASV level. (B) Venn diagram of ASVs distribution among the two types. (C), (D) and (E) The Sobs indices, Pd indices and Shannon indices of the two types. Differences were assessed by Mann–Whitney U tests and are denoted as ∗p < 0.05, ∗∗p < 0.01 and ∗∗∗p < 0.001.

The major phyla of captive striped hamster gut microbiota were Firmicutes, Bacteroidota and Actinobacteriota, while at the family level, Muribaculaceae, Lactobacillaceae, Lachnospiraceae, Erysipelotrichaceae and Bacillaceae were the major members (Fig. S3). LEfSe results (LDA >3, p <0.05) showed that there was a significant enrichment of many bacterial taxa, such as family Lactobacillaceae, family Spirochaetaceae, genus Lactobacillus, genus Treponema and genus Pygmaiobacter in Type 1, while the relative abundances of the order Clostridia_UCG-014, family Hungateiclostridiaceae, and genera Turicibacter and Ruminiclostridium were higher in Type 2 (Fig. 2A). The area under the ROC curve (AUC value) based on the top 30 ASVs was 0.74, indicating that differences in these bacteria could effectively distinguish the gut microbial types (Fig. 2B). Among the top 30 ASVs, six showed significantly different levels of relative abundance between the two types; Type 1 had a higher relative abundance of ASV1 (genus Lactobacillus) (p = 0.002), ASV616 (class Bacilli) (p < 0.001), ASV140 (species Lactobacillus vaginalis) (p = 0.015) and ASV101 (species Lactobacillus vaginalis) (p = 0.013) than Type 2, while Type 2 had a higher relative abundance of ASV2 (genus Turicibacter) (p = 0.002) and ASV788 (family Muribaculaceae) (p = 0.009) than Type 1 (Fig. 2C).

Figure 2 Variations in gut microbiota between the two gut microbial types.

(A) LEfSe identification of gut microbial taxa with significant differences (LDA > 3, p < 0.05). (B) ROC curve calculated by the top 30 ASVs, and the area under the ROC curve (AUC) and 95% confidence intervals are also shown. (C) Six of 30 ASVs that showed significantly different relative abundances between the two types. Differences were assessed by Mann–Whitney U tests and are denoted as ∗p < 0.05, ∗∗p < 0.01 and ∗∗∗p < 0.001.

Co-occurrence networks and functional differences of the two gut microbial types

As shown in Fig. 3 and Table S1, by using the relative abundance of the top 30 ASVs to calculate and display the co-occurrence networks, we found that the numbers of total links were 25 and 36, and the average degrees were 2.381 and 3.273 in Type 1 and Type 2, respectively. In addition, the average clustering coefficient and the number of total triangles in the network of Type 1 (0.397; 8) were all less than those of Type 2 (0.563; 25). Notably, in Type 1, the node that had the greatest degree belonged to the genus Ruminococcus, while the nodes with the greatest degree in the network of Type 2 belonged to the genera Lactobacillus and Bacillus.

Figure 3 Co-occurrence networks of the gut microbiota calculated by the top 30 ASVs.

Nodes represent each ASV, and their sizes indicate different degrees. Links represent significant (p < 0.05) and strong (Spearman’s correlation greater than 0.6 or lower than −0.6) correlations (green dotted lines: negative; red solid lines: positive).

Through PCoA based on the Bray–Curtis distances of the KOs, we found that there was a separation between the two gut microbial types, and a pairwise Adonis test revealed functional characteristics to be significantly different (R2 = 0.153, p = 0.025) (Fig. 4A). By using STAMP to verify the significant differences in level-2 KEGG pathways, we found that the abundances of genes involved in pathways such as xenobiotics biodegradation and metabolism, membrane transport and carbohydrate metabolism were enriched in Type 1, while genes involved in the pathways of amino acid metabolism and metabolism of cofactors and vitamins were enriched in gut microbial Type 2 (Fig. 4B).

Figure 4 Differences in gut microbial functions between the two gut microbial types.

(A) PCoA based on Bray–Curtis distances of KOs. (B) Significant differences in level-2 KEGG pathways between the two types (Welch t test, p < 0.05).

The role of sex in dividing gut microbial types and the sex-specific differences in bacterial taxa

Among the male individuals, five were assigned to Type 1, while three were assigned to Type 2; the female individuals harbored nine Type 1 members and seven Type 2 members (Fig. 5A). Fisher’s exact test showed that the sex of hosts did not influence the classification of gut microbial types (p = 1.000). The AUC value based on the top 30 ASVs was 0.5, indicating that these bacteria could not effectively distinguish individuals of different sexes (Fig. 5B), and there was no significant difference in the relative abundances of these ASVs between male and female hamsters. Additionally, there were no significant differences in the Sobs index (p = 0.500), Pd index (p = 0.257) and Shannon index (p = 0.926) between male and female samples, indicating that the alpha diversity of the gut microbiota in male and female hamsters was similar (Fig. 5C). Male and female samples were not separated in PCoA calculated using ASVs based on Bray–Curtis, unweighted UniFrac and weighted UniFrac distances, and the results of Adonis tests ( R2 = 0.037, p = 0.722; R2 = 0.049, p = 0.218; R2 = 0.041, p = 0.394) also supported this result (Figs. 5D–5F).

Figure 5 The role of sex in the differentiation of gut microbial types.

(A) Distribution of male and female individuals in gut microbial types; differences were assessed by Fisher’s exact test. (B) ROC curve calculated by the top 30 ASVs, and the area under the ROC curve (AUC) and 95% confidence intervals are also shown. (C) The Sobs indices, Pd indices and Shannon indices of males and females. Differences were assessed by Mann–Whitney U tests and are denoted as ∗p < 0.05, ∗∗p < 0.01 and ∗∗∗p < 0.001. (D), (E) and (F) PCoA based on Bray–Curtis, unweighted UniFrac and weighted UniFrac distances calculated using ASVs.

Although there were no significant differences in alpha diversity between male and female individuals, we did detect differences in the relative abundances of particular taxonomic groups. LEfSe results (LDA >2) showed that over all the samples, there was a higher relative abundance of members of the genera Alistipes and Odoribacter in males compared with females, while the relative abundances of family Tannerellaceae and genus Parabacteroides were higher in females (Fig. 6A). Within Type 1 or Type 2, male individuals also had higher relative abundances of family Rikenellaceae, genus Alistipes, family Marinifilaceae and genus Odoribacter (Figs. 6B and 6C). However, there were some bacteria for which the sex difference in relative abundance was not consistent among the two types. For example, higher relative abundances of the genera Sphaerochaeta and Adlercreutzia in males and greater numbers of norank_f__Erysipelotrichaceae and Prevotellaceae_UCG-001 in females were observed only in Type 1 but not in Type 2 (Figs. 6B and 6C).

Figure 6 Sex-specific differences of gut microbiota in captive striped hamsters.

LEfSe identification of gut microbial taxa with significant sex-specific differences in all individuals (A), Type 1 (B) and Type 2 (C) (LDA > 2, p < 0.05).

Discussion

In this study, two gut microbial types combined with different bacterial compositions and functions were identified from captive striped hamsters living in the same laboratory environment. Previous definitions of gut microbial types in wild animals were more understood as a passive reaction process to different external interferences (Ding & Schloss, 2014; Wang et al., 2014; Yu, Li & Li, 2022; Liu et al., 2022), and these responses will help hosts better adapt to various environmental characteristics. The significant environmental variation from wild to captive conditions will greatly alter the animals’ original gut microbiota, and dietary changes can significantly disrupt the previous clustering state of the bacterial community (Schmidt, Mykytczuk & Schulte-Hostedde, 2018; Liu et al., 2019). However, the influence of the wild environment still needs to be considered in gut microbial typing in this experiment because these communities might be different ecological states that exist in nature and further developed under captive circumstances. Captivity is stressful for wild hosts, but the responses to this stress are usually characterized by individual differences, and the gut microbiota of animals will also be prone to diverge in different directions under uniform stimuli (Diaz & Reese, 2021), which can be accompanied by potential impacts of the Anna Karenina effect (Zaneveld, McMinds & Vega Thurber, 2017). Stochastic processes in community assembly play an important role in shaping animals’ gut microbiota in both wild and captive environments, although their effect is relatively weaker in controlled environmental conditions (Bittleston et al., 2020; Li et al., 2022). This could result in the merging or redifferentiation of gut microbiota for forming functionally distinct microbial communities, indicating that the two types in captive striped hamsters were different peaks on an optimization landscape, implying that there should be more potential types in larger surveys. To clarify the role of the above factors in gut microbial typing, more complex experimental designs combined with different time points and control groups are needed in future studies. Biologically speaking, multiple types presented in the gut microbiota can be beneficial for ensuring population tolerance to potential complex external stress. As mentioned earlier, the striped hamster is a widely distributed rodent (Xue et al., 2021), and our results can partly explain its high adaptability to the environment from the perspective of host-gut microbe interactions.

The number of ASVs that were shared among two gut microbial types was much greater than that unique to each type, showing variations in the distribution of bacterial communities between the two types while proving that intraspecific individual differences are limited to a certain range and usually far less than interspecific differences (Song et al., 2020). There was a significant difference only in Shannon indices between the two gut microbial types, indicating that they were characterized by overall similarity but a distinct bacterial composition, which are all normal modes resulting from differentiation of gut bacteria. Type 2 may be slightly better in terms of competitiveness, as a higher alpha diversity of the gut microbiome reflects that the internal ecosystem is more resilient to perturbations (Lozupone et al., 2012). Furthermore, the complexity and cohesion of the gut microbial co-occurrence network can affect their responses to environmental stress (Riera & Baldo, 2020; Hernandez et al., 2021). We predict that the network of Type 1 was easily changed due to external factors, but it also easily recovered, while a complex network of Type 2 was more resistant to pressures but did not recover as easily. These results indicate that the two types of gut microbiota may have different strategies for maintaining community stability.

Firmicutes and Bacteroidota were identified as the two most dominant phyla in the gut microbiota of striped hamsters. These two bacterial phyla are usually the major components of the gut microbial community in mammals, and Firmicutes is vital for the digestion and fermentation of fiber, while Bacteroidota promotes the digestion of fat and polysaccharides (Chen et al., 2016). However, the relative abundances of many bacterial taxa were inconsistent among the two gut microbial types, especially at low classification levels, and the dominant bacterial taxa of each gut microbial type were also different from those of humans, rats, and insects (Ding & Schloss, 2014; Li et al., 2015; Zhang et al., 2019). The genus Lactobacillus belongs to the family Lachnospiraceae and is a potential beneficial gut bacterial taxon and the main producer of short-chain fatty acids, participating in various complex metabolic processes (Vacca et al., 2020). The genus Treponema belongs to the family Spirochaetaceae and is usually associated with the ability to breakdown dietary fibers (Kawasaki et al., 2020; Zhao et al., 2023). The enrichment of these bacteria led to a higher carbohydrate metabolism capacity in the Type 1 community. Bacteria of the class Clostridia (e.g., order Clostridia_UCG-014 and genus Ruminiclostridium) are involved in protein metabolism (Zhao et al., 2019); thus, Type 2 had a higher amino acid metabolism capacity. The individuals of different types were significantly separated in PCoA using the Bray–Curtis distance metric of KOs, proving the variations in gut microbial function between the two gut microbial types. Striped hamsters inhabit different farmlands where various crops are grown, and individuals with a Type 1 gut microbiota may be inclined to choose high carbohydrate food resources such as corn and wheat, while the Type 2 gut microbiota may drive its hosts to live in soybean or peanut fields that have rich protein resources. These differences help prevent excessive concentrations of hamsters in narrow areas, thereby reducing the pressure of intraspecific competition.

We previously predicted that individuals of the same sex may cluster together and have a certain consistency with the classification of gut microbial types. However, the distribution of individuals with different sexes in gut microbial types was not significantly biased. Moreover, male and female samples were not separated in dimension reduction sorting. It should be acknowledged that the sample size of hamster participants in this experiment is relatively small, which may impose certain limitations on the exploration of sex-specific differences in gut microbiota. This issue was also mentioned in a study of wild Brandt’s voles (Lasiopodomys brandtii), which found no variation in beta diversity between females and males (Xu & Zhang, 2021), while another study involving hundreds of human volunteers discovered both significant separation of overall gut microbial structure in PCoA analysis and variations in specific bacterial taxa (Takagi et al., 2019). Since the gut microbiota was segregated into two distinct types, the potential sex-related difference might be even harder to detect under a more powerful classification framework. Taxa that rank high in relative abundance are usually defined as core bacteria and are regarded as the key to variations in gut microbiota (Cernava et al., 2019; Ren et al., 2021), but there was no significant difference in the relative abundances of the top 30 ASVs between male and female striped hamsters, which might be due to the environmental pressure of indoor breeding. Biological sex can affect gut microbiota through distinct host physiological indicators such as hormones and by making hosts have different personalities and lifestyles (Valeri & Endres, 2021), and captivity will hinder the latter to a certain extent by eliminating environmental interference while making the animals feel pressure after lifestyle disruption (Diaz & Reese, 2021). Therefore, studies on artificially raised rodents have found similar results: sex bias of some gut microbial taxa, without the separation of overall structure in dimension reduction sorting (Org et al., 2016; He et al., 2021). Interestingly, the variations between male and female hamsters showed inconsistency across the two types, especially the bacterial taxa enriched in females, such as genera norank_f__Erysipelotrichaceae and Prevotellaceae_UCG-001, which were found only in Type 1, suggesting that sex could shape the gut microbiome differently in striped hamsters within the context of the very different community types.

Conclusions

In summary, this study found that the gut microbiota of captive striped hamsters was separated into two distinct types with different compositions and functions: enrichment of the genera Lactobacillus, Treponema and Pygmaiobacter in one gut microbial type and enrichment of the genera Turicibacter and Ruminiclostridium in the other. The former type had higher carbohydrate metabolism ability, while the latter harbored a more complex co-occurrence network and higher amino acid metabolism ability. This suggests that gut microbial types are not just the related responses of individuals living in different environments, as type differentiation can still appear in normal animals’ gut microbiota under the same conditions. However, the distribution of individuals of different sexes in gut microbial types was not significantly biased. Sex-specific differences were inconsistent with intertype differences and had different manifestations in distinct types.

Supplemental Information

Supplemental Information 1 Rarefaction curves of fecal samples

Click here for additional data file.

Supplemental Information 2 Calinski-Harabasz (CH) indices for optimal numbers of gut microbial types

Click here for additional data file.

Supplemental Information 3 Composition of gut bacteria at the phylum and family levels

Click here for additional data file.

Supplemental Information 4 Network indices of the two gut microbial types

Click here for additional data file.

Supplemental Information 5 The mapping file and ASV table

Click here for additional data file.

We would like to thank Hao Zhang, Xingchen Wang and Mingyang Jia for their help in the experiment.

Additional Information and Declarations

Competing Interests

Author Contributions

Animal Ethics

Data Availability

The authors declare there are no competing interests.

Chao Fan conceived and designed the experiments, performed the experiments, analyzed the data, prepared figures and/or tables, authored or reviewed drafts of the article, and approved the final draft.

Yunjiao Zheng performed the experiments, authored or reviewed drafts of the article, and approved the final draft.

Huiliang Xue performed the experiments, authored or reviewed drafts of the article, and approved the final draft.

Jinhui Xu conceived and designed the experiments, analyzed the data, prepared figures and/or tables, authored or reviewed drafts of the article, and approved the final draft.

Ming Wu analyzed the data, authored or reviewed drafts of the article, and approved the final draft.

Lei Chen analyzed the data, authored or reviewed drafts of the article, and approved the final draft.

Laixiang Xu conceived and designed the experiments, authored or reviewed drafts of the article, and approved the final draft.

The following information was supplied relating to ethical approvals (i.e., approving body and any reference numbers):

All procedures were approved by the Biomedical Ethics Committee of Qufu Normal University (dwsc2023005).

The following information was supplied regarding data availability:

The raw reads are available at the National Center for Biotechnology Information: PRJNA937404.

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
