# Peer review of "Different gut microbial types were found in captive striped hamsters"

_PeerJ, doi:10.7717/peerj.16365_

## Round 0.1 · original submission · Major Revisions

The manuscript needs major revisions, especially for experimental design, sample size, and time point of sample collection. Further, the comments raised by the reviewers should be keenly addressed in the revised manuscript.

Reviewer 1 ·

Basic reporting

The manuscript by Fan et al is well-written, and structured in accordance with PeerJ guidelines.

The cited literature, while relevant to the topic on the surface, sometimes concerns animal species that are not closely related to the subject (striped hamsters). The authors could consider trying to focus more on rodents and more closely related species, so it is a bit more coherent and relevant to the topic of study.
Overall background and context is okay.

The presented results are all relevant to the research question posed.

I did not see a data availability statement in the manuscript, this should be added.

Experimental design

The authors pose the research question whether or not sex contributes to gut microbial community typing in striped hamsters. At the same time they also mention that little knowledge exists regarding the microbial communities of this species. I think it is a missed opportunity that the microbial community description is quite limited (top 30 ASV only), and the paper could be improved by adding information regarding the microbial community composition between sexes, as well as the typing analysis, seeing as this is the topic of the study.

The experimental setup worries me. Hamsters were trapped in the wild, and then acclimatised for 2 months prior to sample collection. During this period of time they were all kept under laboratory conditions and fed commercial foods. This radical change in environment, food and social structure (animals were kept alone) will very likely have caused stress which also impacts the microbiome, and could add additional noise to the results. Also, I wonder if 2 months is enough to adapt the microbiome fully to the new food source, compared to the wild.
In addition, the sampling size is small, and most worryingly, uneven (8 vs 16 individuals of both sexes). This could potentially skew the analysis as it is not sure that this number of individuals is enough to have a robust dataset that covers all individual variation, and the variation from the females may have had a greater effect compared to the males as there were 2x as many. As the microbial community between sexes is not explored in detail, it is difficult to assess if this is the case in the presented study.
These factors should be taking into consideration when (re-)analysing the results, and more conservative observations may need to be made.

Methods are described well, with minor missing details: No reference given for the primer set used, missing information regarding how diversity indices were calculated.

Validity of the findings

The data from the study has not been made available in a repository, this needs to be added.

The general statistical treatment of the data was robust and statistically sound. I am however missing information regarding the quality of the sequences obtained from the 16S rRNA gene analysis. It would be good to add this so the reader can understand the data that the analysis was based on more fully.

Based on my concerns regarding the experimental design, I am not sure that the data presented supports the conclusions drawn. I am worried that the data contains too much noise to be able to answer the research question posed. Adding a detailed analysis of the microbial community data with focus on the sex of the analysed species will allow the authors to better assess their conclusion that sex has no influence on microbial typing (and the microbiota in general).

Additional comments

I appreciate the submission of this manuscript where no difference was found. This is just as important as reporting a difference or a "result", but is often forgotten, and I wish this was a more standard practice.

Reviewer 2 ·

Basic reporting

The literature references on sex and the microbiome are very lacking. The authors do a very cursory job of discussing the potential role of sex in shaping the mammalian gut microbiome. They only cite three papers in this regard and one on ticks that doesn’t seem at all relevant. The literature on sex differentiation and the mammalian microbiome is large and growing, and there are many studies with much larger sample sizes than the present study that have identified sex related differentiation in the gut microbiome. Here are two recent reviews on this topic.

https://www.frontiersin.org/articles/10.3389/fmicb.2021.711137/full

https://rep.bioscientifica.com/view/journals/rep/165/2/REP-22-0303.xml

The authors need to increase their references and understanding of how sex may affect the microbiome. It is an important factor, but not the most important (top 10). I go into more detail about this the pdf. But sex should be a minor focus of this paper.

Experimental design

This is a descriptive study and it fine in that regard. The biggest problem is that the authors only sampled at the end of the study, so they cannot know if the Types they report developed spontaneously as they claim. However, the basic study of the gut microbiome in the hamsters is well done and I ask them to provide the metadata and code so their results can be more useful and reproducible.

They cannot fix this at this point, but they should modify their claims about spontaneous developments of type that might already exist in nature.

Validity of the findings

The strong separation of microbial consortia in this hamsters is very interesting and should be published. However, how these developed and why are unknown and can only be suggested. The conclusions should reflect this and the fact that there may be more types out there in the wild.

Additional comments

See pdf.

Annotated reviews are not available for download in order to protect the identity of reviewers who chose to remain anonymous.

---

## Round 0.2 · Minor Revisions

The manuscript has been significantly improved. The conclusion of the study should be in light of the results obtained. The manuscript still needs English language proofreading.

**Language Note:** The Academic Editor has identified that the English language must be improved. PeerJ can provide language editing services - please contact us at [email protected] for pricing (be sure to provide your manuscript number and title). Alternatively, you should make your own arrangements to improve the language quality and provide details in your response letter. – PeerJ Staff

Reviewer 1 ·

Basic reporting

The revised manuscript by Fan et al has shifted focus to a more conservative and broad description of the gut microbial types observed in striped hamsters after 2 months captivity. The authors have put in a lot of work to address all review comments, and I feel that the manuscript is much improved by the revisions.
The addition of the descriptive results add value to the manuscript, and now provides better insight into the gut microbial community findings.
The literature cited in the manuscript now better supports the text in both the introduction and discussion section of the paper.

The only comment I have here is that the English used in the new additions is not always entirely grammatically correct, and reads awkwardly in some places. I recommend that the authors polish their text one more time.

Experimental design

No comment. My concerns about the experimental design and presented data have been addressed in the revised manuscript.

Validity of the findings

The data storage information has been added, and with the addition of the new descriptive results, I think the manuscript is much improved. I have no further comments here.

Additional comments

No comment.

Reviewer 2 ·

Basic reporting

no comment

Experimental design

no comment

Validity of the findings

no comment

Additional comments

Comments on revision:
The authors have made a lot of positive changes to the manuscript esp. by including the metadata availability and link to the raw sequences, and the code used for analysis. They also changed the focus to the types which is very interesting and acknowledged the issues with the model interpretation.
I am still having issues with their description of the effects of sex on the stripped hamster microbiome. Their conclusions in the abstract are contradicted by their own data and confusing. For example, this sentence in the abstract contradicts their findings: “We did not detect that sex could dominant the classification of gut microbial types, but the variations between male and female individuals were inconsistency across the two types.” In fact, the authors DID find an effect of sex in terms of the relative abundance of taxa. Here is what they say in the Discussion: " suggesting that sex could shape the gut microbiome variously in striped hamsters within the context of the very different community types.”
This passage from the results, which I edited (please check the taxonomic names), reveals that there were detectable sex effects in the microbiome: “Although there were no significant differences in alpha diversity between male and female individuals, we did detect differences in the relative abundances of particular taxonomic groups. LEfSe results (LDA > 2) showed that over all the samples there was a higher relative abundance of members of the genera Alistipes and Odoribacter in males compared with females, while the relative abundances of family Tannerellaceae and genus Parabacteroides were higher in females (Fig. 6A). Within Type 1 or Type 2, male individuals also had a higher relative abundances of family Rikenellaceae, genus Alistipes, family Marinifilaceae and genus Odoribacter (Figs. 6B and C). However, there were some bacteria for which the sex difference in relative abundance was not consistent among the two types. For example, higher relative abundances of the genera Sphaerochaeta and Adlercreutzia in males, and greater numbers of norank_f__Erysipelotrichaceae and Prevotellaceae_UCG-001 in females were only observed in Type 1, but not in Type 2 (Figs. 6B and C).
What I think authors are trying to say is that the Types they found were not associated with Sex, but it sounds like they are saying that sex had no effect on the microbiome. It did have an effect, but it was just more subtle.
Thus I ask the authors to change their abstract to read the following.
Change: “We did not detect that sex could dominant the classification of gut microbial types, but the variations between male and female individuals were inconsistency across the two types.”
To something like: “The gut microbial types were not associated with sex, however we did find sex differences in the relative abundances of certain bacterial taxa including some type-specific sex differences.”
Other edits:
1) See my changes to the Results paragraph above.
2) Change: “Actually, stochastic processes in community assembly play an important role in
shaping animals’ gut microbiota in both wild and captive environments, though their proportion is relatively less in the latter condition (Bittleston et al., 2020; Li et al., 2022).

To: “Stochastic processes in community assembly play an important role in shaping animals’ gut microbiota in both wild and captive environments, though their effect is relatively less in controlled environmental conditions (Bittleston et al., 2020; Li et al., 2022).”
3) The authors use “researches” when they mean “research” (this word is singular and plural) in several places.

---

## Round 0.3 · accepted · Accept

I feel happy to accept the manuscript for publication.

Reviewer 1 ·

Basic reporting

No comment.

Experimental design

No comment.

Validity of the findings

No comment.

Additional comments

No comment.